# Prediction of Gastrointestinal Tract Cancers Using Longitudinal Electronic Health Record Data

**DOI:** 10.3390/cancers15051399

**Published:** 2023-02-22

**Authors:** Andrew J. Read, Wenjing Zhou, Sameer D. Saini, Ji Zhu, Akbar K. Waljee

**Affiliations:** 1Division of Gastroenterology and Hepatology, Department of Internal Medicine, University of Michigan, Ann Arbor, MI 48109, USA; 2Institute for Healthcare Policy and Innovation, University of Michigan, Ann Arbor, MI 48109, USA; 3Michigan Integrated Center for Health Analytics and Medical Prediction, University of Michigan, Ann Arbor, MI 48109, USA; 4Department of Statistics, University of Michigan, Ann Arbor, MI 48109, USA; 5VA HSR&D Center for Clinical Management Research, Ann Arbor, MI 48105, USA

**Keywords:** gastrointestinal cancers, prediction model, machine learning

## Abstract

**Simple Summary:**

Cancers of the gastrointestinal tract—including the esophagus, stomach, and intestines—are often diagnosed at an advanced stage, when curative treatments are rare. These cancers can all cause gastrointestinal bleeding, but this often occurs gradually and may be unnoticed by patients. Changes in routine laboratory parameters such as the complete blood count may be able to show these subtle changes prior to clinical presentation or the development of iron deficiency anemia. The aim of our study was to develop models for the prediction of luminal gastrointestinal tract cancers (esophageal, gastric, small bowel, colorectal, anal) using data routinely available within an electronic health record, in a retrospective cohort from an academic medical center. The cohort included 148,158 individuals, with 1025 gastrointestinal tract cancers. We found that longitudinal prediction models using the complete blood count outperformed a single timepoint logistic model for 3-year cancer prediction.

**Abstract:**

Background: Luminal gastrointestinal (GI) tract cancers, including esophageal, gastric, small bowel, colorectal, and anal cancers, are often diagnosed at late stages. These tumors can cause gradual GI bleeding, which may be unrecognized but detectable by subtle laboratory changes. Our aim was to develop models to predict luminal GI tract cancers using laboratory studies and patient characteristics using logistic regression and random forest machine learning methods. Methods: The study was a single-center, retrospective cohort at an academic medical center, with enrollment between 2004–2013 and with follow-up until 2018, who had at least two complete blood counts (CBCs). The primary outcome was the diagnosis of GI tract cancer. Prediction models were developed using multivariable single timepoint logistic regression, longitudinal logistic regression, and random forest machine learning. Results: The cohort included 148,158 individuals, with 1025 GI tract cancers. For 3-year prediction of GI tract cancers, the longitudinal random forest model performed the best, with an area under the receiver operator curve (AuROC) of 0.750 (95% CI 0.729–0.771) and Brier score of 0.116, compared to the longitudinal logistic regression model, with an AuROC of 0.735 (95% CI 0.713–0.757) and Brier score of 0.205. Conclusions: Prediction models incorporating longitudinal features of the CBC outperformed the single timepoint logistic regression models at 3-years, with a trend toward improved accuracy of prediction using a random forest machine learning model compared to a longitudinal logistic regression model.

## 1. Introduction

Malignancies of the gastrointestinal (GI) tract—including esophageal, gastric, small bowel, colorectal, and anal cancers—are a leading cause of morbidity and mortality, with over 200,000 new diagnoses and approximately 80,000 deaths per year in the United States [1]. While routine screening is recommended for colorectal cancer (CRC), many patients go unscreened, particularly in vulnerable and underserved populations [2]. Recent studies have also noted a rising incidence of CRC in younger patients for whom screening may be impractical or ineffective [3,4,5,6]. As a result, even with lowering the age for initiation of CRC screening to 45 [7,8], existing screening programs for GI tract cancers remain inadequate, and there is no routine screening recommended for GI tract cancers in average-risk adults other than for CRC (e.g., stool testing or colonoscopy).

As GI tract cancers often do not present clinically until they are at an advanced stage, early diagnosis is critical for improving outcomes [9,10,11,12,13]. Improved diagnosis could be achieved by leveraging a physiological common link in luminal GI tract cancers: gradual occult blood loss, ultimately resulting in iron deficiency anemia (IDA) [14]. Healthcare providers often obtain complete blood counts (CBCs) as part of routine clinical care [15,16], but clinicians do not always diagnose IDA accurately and may not obtain the recommended diagnostic evaluation of bidirectional endoscopy (esophagogastroduodenoscopy [EGD] and colonoscopy) for patients with new-onset IDA [15,17,18]. One approach that has been described is the use of electronic trigger tools based on concerning patterns in laboratory data such as new-onset IDA [19]; however, they have not been widely adopted in clinical practice. In addition, site-specific prediction models have examined the association between longitudinal changes in CBCs and the diagnosis of CRC [20,21,22], but new models for prediction of occult malignancy within the entire GI tract are needed. Such models could utilize existing longitudinal laboratory data combined with other patient characteristics stored within the electronic health record (EHR) and serve as automated tools to help improve diagnosis.

In this paper, we describe the development of models for the prediction of luminal GI tract cancers (esophageal, gastric, small bowel, colorectal, anal) using a single-center retrospective cohort. We developed and compared models using single timepoint logistic regression, longitudinal logistic regression, and longitudinal random forest machine learning.

## 2. Materials and Methods

The study was conducted as a single-center, retrospective cohort study of patients receiving care at an academic medical center (Michigan Medicine, Ann Arbor, MI, USA) between 2004–2018. This study was approved with a waiver of informed consent by the University of Michigan Institutional Review Board (HUM00156237), due to the large retrospective nature of the study. Data analysis and model development were performed using SAS 9.4 (SAS Institute, Cary, NC, USA) and Python 3.8 (Python Software Foundation, Wilmington, DE, USA). Elements of the TRIPOD guidelines for transparent reporting of multivariable prediction models were used [23].

Subjects were identified as individuals from the Michigan Medicine Clinical Data Warehouse who had at least 2 CBCs within a rolling 2-year time frame between 1 January 2004 and 31 December 2013. Michigan Medicine is a large referral center as well as a primary care system. We used the presence of 2 CBCs to identify patients seeking regular care at Michigan Medicine and to provide at least two data points for a longitudinal prediction model. Subjects were excluded if age < 18, given the low incidence of GI tract cancers and paucity of routine blood draws in a pediatric population. Data were collected from the date of subject’s inclusion until 31 December 2018 (or diagnosis of GI tract cancer), including laboratory values from complete blood counts (CBCs), basic metabolic panels (BMPs), age, sex, self-reported race (as documented in the EHR demographics field), and Body Mass Index (BMI, in kg/m²). Data pre-processing was performed in SAS 9.4 (SAS Institute, Cary, NC, USA), with merging of the demographic, laboratory, biometric, and cancer registry data into a unified file. Biologically implausible laboratory or BMI values were excluded.

### 2.1. Predictor Variables

Each model included patients’ demographic variables (age, sex, race), BMI, the individual component variables of the CBC, and the most recent BMP components. We included all the variables from the CBC since subtle changes within laboratory parameters other than hemoglobin or hematocrit may also reflect an iron-deficient state, e.g., elevated red cell distribution width (RDW), low mean cellular hemoglobin (MCH), and low mean cellular volume (MCV) [24,25]. We also included the BMP in these models, which may reflect comorbidities with associated potential links to GI tract cancers, e.g., reported associations between CRC and chronic kidney disease [26] (suggested by elevated blood urea nitrogen and creatinine) and gastric cancer and diabetes [27,28,29] (as might be suggested by hyperglycemia). As machine learning methods can identify patterns that are imperceptible to clinicians, we included all variables from the CBC and BMP, as these methods tend to perform better with additional data rather than making fixed assumptions about the importance of individual predictors.

### 2.2. Primary Outcome

The primary outcome was the diagnosis of a GI tract cancer, as determined by linkage to the University of Michigan Cancer Center Registry, which contains confirmed pathologic diagnoses of all cancers diagnosed at Michigan Medicine. We chose this method due to the lack of specificity of International Classification of Diseases (ICD)-9/10 codes at differentiating between the time of a diagnosis and the time of documentation in a chart (e.g., potential that a newly charted diagnostic code may reflect documentation of an existing GI tract cancer that occurred many years prior rather than a new diagnosis of cancer). In addition, during the study period, Michigan Medicine updated its EHR system (beginning in 2012), which resulted in overlapping usage of ICD9 and ICD10 codes beginning in 2012. As a result, we selected the cancer registry as it provided a consistent source of confirmed cancers.

We limited the outcomes to the diagnosis of luminal GI tract cancers, as defined for this study as cancers of the esophagus, stomach, small intestine, colon, rectum, or anus. Non-luminal GI tract cancers such as pancreaticobiliary cancers or liver cancers were excluded from this analysis as we were primarily interested in potential effects of occult GI tract bleeding, as might be reflected by changes in the CBC. For individuals with GI tract cancers within the cancer registry, we used the date of the diagnosis as the individual’s final observation. For individuals with no GI tract cancer within the registry, we used the date of the last recorded CBC to define the end of the observation period.

### 2.3. Model Development

We used three techniques of prediction model development: (1) single timepoint multivariate logistic regression; (2) multivariate logistic regression incorporating summarized longitudinal features; and (3) random forest machine learning incorporating longitudinal features. For each prediction technique, we developed a prediction model for diagnosis of a GI tract cancer at 6-months, 1-year, 3-years, and 5-years. The eligible sub-population for each time interval was determined in SAS and exported to Python for model building. For each model prediction interval, subjects were included who had at least 2 CBCs prior to the prediction interval, and at least one CBC within the year prior to the beginning of the prediction interval. For example, for the 1-year prediction interval, subjects were included who had at least one CBC between 1 and 2 years prior to the final observation (diagnosis of GI tract cancer/no cancer). For the 6-month prediction, we included those subjects who had at least one CBC between 6 and 12 months prior to the final observation.

For the single timepoint multivariate logistic regression prediction models, we selected observations on the date of the CBC laboratory draw that was closest to the prediction window. Predictor variables included: age, sex, race, most recent BMI, values from the individual components of the CBC on that date (Hemoglobin [Hgb], platelets [Plt], White Blood Cell [WBC] count, etc.), and the values from the most recently available BMP (Sodium [SOD], Glucose [Gluc], blood urea nitrogen [UN], creatinine [Creat], etc.).

To incorporate longitudinal elements into the logistic regression and random forest machine learning models, we calculated summary statistics for each subject, summarizing the trends of the individual components of the CBC in the 3 years prior to the prediction window. For example, for each individual component of the CBC (Hgb, Plt, WBC, etc.), we summarized the values over the prior 3 years by the maximum and minimum; the maximum and minimum slope of each predictor variable (i.e., where the slope is the ratio of the change in value/difference in time between two consecutive observations); and the total variation (mean of the absolute value of the slopes). Because the laboratory data were obtained through routine clinical care (irregular intervals that varied between individuals), the use of slopes between observations helped to better describe changes in laboratory values over time. These summary statistics were then added to the predictor variables in the base single timepoint logistic regression models for the longitudinal logistic regression and longitudinal random forest machine learning models. Missing values for individual summary statistics or individual laboratory parameters were determined through imputation using the median value observed across the cohort.

### 2.4. Statistical Analysis

We calculated descriptive statistics of the cohort at baseline inclusion. For each prediction interval (6-months, 1-year, 3-years, 5-years), we performed a random 70/30 split, with 70% of the individuals in a training set and 30% in a testing set. Within each prediction interval, we used a training set to fit single timepoint logistic regression, longitudinal logistic regression, and longitudinal random forest machine learning models and evaluated prediction performances using the same testing set. We repeated this procedure 10 times and reported the mean performance characteristics on the testing set over 10 random splits. We implemented logistic regression models with L2 regularization to minimize the potential effects of overfitting. To tune the optimal penalty coefficient for regularized logistic regression, we conducted 5-fold cross-validation, and then the model was fitted with the selected coefficient using the training set.

For the longitudinal machine learning model, we used the random forest technique. Random forest machine learning is an ensemble, tree-based machine learning algorithm used to classify individuals [30,31], which has been used in multiple prior models and described in detail [32,33,34]. Briefly, each tree classifies the individuals independently. Next, the random forest combines the decisions from each tree to generate a final classification for an individual, which can be understood as the majority vote from trees. We also used 5-fold cross-validation to tune the hyperparameters related to the number, size, and feature of trees in the random forest. For both logistic regression and random forest models, we adjusted the class weight using a built-in argument in the Python scikit-learn package to solve the problem of imbalanced classification (rare events of cancers relative to the population).

Finally, for each model, we determined the area under the receiver operator curve (AuROC), Brier score (measurement of overall performance), and the optimal (maximal) sensitivity/specificity using the test dataset. To balance the sensitivity and specificity, we determined the optimal cut-point, defined here as the point closest to the perfect classification point (0, 1) on the receiver operator curve. We also determined the relative variable importance rankings for the predictor variables in these models. In addition, we performed additional analysis of the performance of the models at predicting cancers by age categories and by GI tract tumor.

## 3. Results

### 3.1. Baseline Cohort

We identified 148,158 individuals who met the inclusion criteria (Table 1). The mean age was 49.4 (SD = 17.3) and the majority were women (62.1%, *n* = 91,995/148,158). Most of the subjects were Caucasian (81.3%, *n* = 120,385/148,158), with 10.5% being African American (*n* = 15,510/148,158), and 4.6% Asian (*n* = 6795/148,158). Within the cohort, we identified 1025 GI tract cancers during the study period: the majority were CRCs (53.5%, *n* = 548/1025), followed by gastric cancer (16.6%, *n* = 170/1025), esophageal cancers (12.5%, *n* = 128/1025), anal cancers (8.6%, *n* = 88/1025), small bowel cancers (7.7%, *n* = 79/1025), and not otherwise specified within the GI tract (1.2%, *n* = 12/1025).

### 3.2. Single Timepoint Prediction Using Logistic Regression

We developed prediction models for the diagnosis of GI tract cancer at 6-months, 1-year, 3-years, and 5-years using multivariate logistic regression at a single timepoint (the last CBC prior to the prediction interval). We included patients’ age, sex, race, BMI, individual components of the CBC, and the most recent BMP (on or prior to the date of the CBC used for prediction). The results of the models’ performance are shown in Table 2. For 6-month prediction of GI tract cancer, the area under the receiver operator curve (AuROC) was 0.697 (95% CI 0.679–0.715), corresponding to a sensitivity of 0.603 and specificity of 0.690 in this population, with a Brier score of 0.007. At increasing time periods of prediction, the AuROC increased; however, the Brier score also increased to above 0.2, indicating lower performing models (Table 3).

### 3.3. Longitudinal Logistic Regression Model

We developed longitudinal logistic prediction models for the diagnosis of GI tract cancer (at 6-months, 1-year, 3-years, 5-years) using the predictor variables from the single timepoint logistic regression model with the addition of summary variables of the longitudinal CBCs (maximum/minimum, total variation, maximum/minimum slopes). Addition of these longitudinal features led to higher AuROCs for prediction at 6-months, 1-year, and 3-years as compared to the corresponding single timepoint logistic regression models (Table 2). For example, the 3-year AuROC was 0.735 (95% CI 0.713–0.757) compared to 0.683 (95% CI 0.665–0.701) for the single timepoint logistic regression prediction model (Figure 1). The 1-year longitudinal logistic regression AuROC was 0.705 (95% CI 0.689–0.722) with a Brier score of 0.008, compared to the 1-year single timepoint logistic regression model of 0.683 (95% CI 0.665–0.701) with a Brier score of 0.224 (indicating poor performance).

### 3.4. Longitudinal Random Forest Machine Learning Model

We developed longitudinal random forest machine learning prediction models for diagnosis of GI tract cancer (at 6-months, 1-year, 3-years, 5-years) using the predictor variables from the single timepoint logistic regression model with the addition of summary variables of the longitudinal CBCs. The random forest model AuROCs were greater than both logistic regression models for 6-months, 1-year, and 3-year predictions (Figure 1), with an AuROC of 0.750 at 3 years (95% CI 0.729–0.771) and a Brier score of 0.116. However, the confidence intervals of the AuROCs overlapped with the longitudinal logistic regression model for all three time periods (Table 2). The variable importance factors for the random forest machine learning models at 1-year and 3-years demonstrated that the most recent (last) mean platelet volume (MPV), minimum MPV, and age were the three most heavily influential variables in these models (Figure 2).

### 3.5. Subanalysis by Age and Tumor Type

We analyzed the longitudinal logistic regression and longitudinal random forest machine learning prediction models for their prediction success at 1- and 3-years by age group and category of GI tract cancer. We selected three age categories: age less than 50, age 50 years or older and less than 75, and age greater than or equal to 75. These ages were selected as they corresponded with screening age groups for colorectal cancer screening during this study period: CRC screening was recommended starting at age 50, not recommended for those less than 50, and individualized screening decision was recommended between ages 75 and 85. There was a trend toward lower AuROCs for those older than 75, suggesting that the models performed less well in this age group, although some of the confidence intervals overlapped, suggesting this was not a statistically significant difference (Table A1).

To describe the imbalanced nature of these groups (overall cohort population was younger, with a median age of 49.4), we determined the imbalance ratio as the ratio of the number of negative samples (individuals without cancer) to the number of positive samples (individuals with cancer) in each category. These findings are consistent with established epidemiological trends of increasing GI tract cancers with age. Although the study was not powered to predict individual GI tract cancers, we calculated the performance of the models on the prediction of individual cancers (Table A2). In this setting, the imbalance ratios were more pronounced, e.g., there were only 30 small bowel cancers with sufficient longitudinal data to make 3-year predictions. The models performed less well in the setting of fewer events for this category.

## 4. Discussion

The results of this retrospective single-center cohort study demonstrate that data within the electronic health record (including CBCs, BMPs, age, sex, race, BMI) can be used to help predict the diagnosis of luminal GI tract cancers (esophageal, gastric, small intestine, colorectal, and anal), with an AuROC of up to 0.750 for prediction of GI tract cancer at 3 years (95% CI 0.729–0.771; Brier score = 0.116) with a random forest machine learning model, compared to the longitudinal logistic regression model with an AuROC of 0.735 (95% CI 0.713–0.757) with a Brier score of 0.205. While there was a trend toward improvement with machine learning compared to longitudinal logistic regression, the overlapping confidence intervals mean the model is not definitively better. One possible explanation is the relative rarity of GI cancers compared to the cohort as a whole and the general need for more events to outperform logistic regression in machine learning techniques [35]. In addition, this lack of superiority of ML has been found in other clinical prediction domains as well, highlighting the strengths of multivariate logistic regression and the difficulties in outperforming these models with newer techniques [34,36]. At 5-years’ prediction, when longitudinal changes would be less likely to have immediate predictive power, the single timepoint logistic regression model had a higher point estimate AuROC than the longitudinal models at 0.703 (95% CI 0.686–0.720), but with a Brier score of 0.213 (indicating overall lower performance). Nonetheless, this study demonstrates important signals that prediction models of luminal GI tract cancers may be useful adjunctive tools to existing clinical intuition and practice guidelines (e.g., that patients with overt GI bleeding or IDA should undergo endoscopic evaluation) [17].

One important aspect of the random forest machine learning method is that predictor variable associations can be identified that may not be otherwise intuitive. For example, mean platelet volume (MPV) was one of the most important variables in the longitudinal random forest machine learning model. There has been growing interest in the potential usefulness of MPV as a marker of systemic inflammation in GI tract cancers, with possible diagnostic implications for gastric and colorectal cancers [37,38,39,40,41] and possible prognostic implications for esophageal cancers [42]. Clinicians rarely use this feature in routine practice, with a prior clinician survey reporting that clinicians consider MPV to be the least useful component of the CBC [24]. Other predictor variables, such as age, are likely more intuitive to clinicians, consistent with established epidemiologic data of increasing incidence of GI tract cancers with increasing age [2,3].

While these models would be inadequate to replace existing CRC [7,8] screening programs [43], they might still have adjunctive utility. For example, guidelines have already implicitly established a tolerance for the “number needed to scope,” or the number of colonoscopies needed to detect one cancer. For example, guidelines recommend bidirectional endoscopy (EGD and colonoscopy) for new-onset IDA [17], with a number needed to scope for a diagnosis of cancer of between to 10 and 100 (incidence ranging between 1 and 10%) [44,45]. Similarly, the threshold for fecal immunochemical testing (FIT) for CRC screening has a PPV of 2.9–7.8% for diagnosis of GI tract cancer [46], corresponding to a number needed to scope of approximately 13 to 35 to diagnose one cancer. Thus, using these reference points, a prediction model utilizing EHR data could be calibrated to achieve a higher specificity, while tolerating a lower sensitivity (as this would not be replacing routine screening), until the positive predictive value reached an acceptable threshold for recommending diagnostic endoscopy. This type of model would be complementary to existing screening programs.

There are several limitations to this study. First, this study was performed retrospectively at an academic medical center, using data collected through routine clinical care, and thus may not apply to other clinical practice settings. The eligible population included all patients receiving care at Michigan Medicine, which includes patients seen by Michigan Medicine primary care providers and specialists. We further narrowed our cohort to individuals with longitudinal follow-up within the Michigan Medicine health system by requiring at least two CBCs over two years. To maximize our eligible cohort, we did not exclude individuals based on the type of provider(s) seen at Michigan Medicine or other exclusions. For this exploratory study, we did not have an external validation cohort, so the accuracy of the models may decline in other settings, as there may be other unmeasured differences between populations. Second, because our inclusion criteria required the presence of at least two CBCs per patient (to determine longitudinal trends), there may be unknown, systematic differences between these patients and those with fewer CBCs (who were excluded from the cohort). An alternate model incorporating a single CBC could have advantages in a setting where primary care follow-up is limited or where CBCs are less commonly obtained. It is also reasonable to consider whether the increased performance of a longitudinal model is “worth” the added computational complexity that would be required for its deployment. Third, for purposes of this analysis, we considered the diagnosis of any GI tract cancer as a binary outcome, given the relatively rare incidence of GI tract cancers relative to the cohort size. However, the tumor biology of GI tract cancers is heterogenous. As a result, there may be different patterns specific for individual subtypes of GI tract cancer that this study was not powered to detect. Models focused solely on a single organ have the potential to have higher specificity but would not be designed to detect other GI tract tumors. Fourth, we were limited by predictor variable inclusion in our models due to a high degree of missingness of some suspected useful variables (CBC with differentials, ferritin), potentially limiting the predictive power relative to prior models for CRC that incorporated CBCs with differentials [20]. An additional limitation is that these models do not incorporate additional clinical history such as the findings of prior EGD or colonoscopy procedures, but we intentionally chose to focus on readily ascertainable parameters from existing EHR data for easier potential use in the future.

## 5. Conclusions

Nonetheless, despite these limitations, this work offers some important contributions to the diagnostic evaluation of GI tract cancers, demonstrating that logistic regression or random forest machine learning models using EHR data can help predict the presence of GI tract cancers. Improved diagnosis in this domain is critical. First, given epidemiologic trends with an increasing incidence of CRC at younger ages, additional detection strategies are needed to identify diseases earlier in this younger cohort, who would not yet have undergone routine CRC screening. Second, given limited endoscopic access in some settings, methods to identify patients at greatest risk of GI tract cancer are increasingly important, as they could help prioritize GI diagnostic evaluations on those individuals at greatest risk. Third, prior work has demonstrated that IDA is not always diagnosed or evaluated fully, meaning that additional automated methods of helping clinicians identify patients at increased risk of GI tract cancers are needed. In summary, these models could help fill an important need and assist clinicians in the diagnosis of GI tract cancers. Further refinement and evaluation of these models in a larger cohort, with external validation, is needed prior to any potential prospective clinical evaluation.

## Figures and Tables

**Figure 1 cancers-15-01399-f001:**
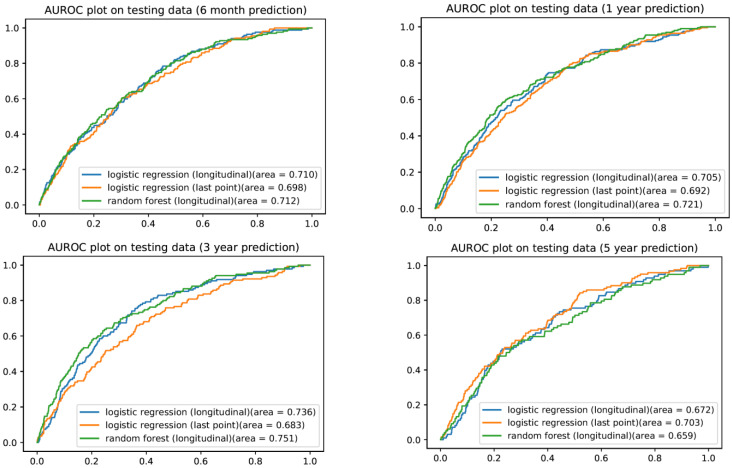
Receiver operator curves for GI tract cancer prediction at 6-months, 1-year, 3-years, and 5-years.

**Figure 2 cancers-15-01399-f002:**
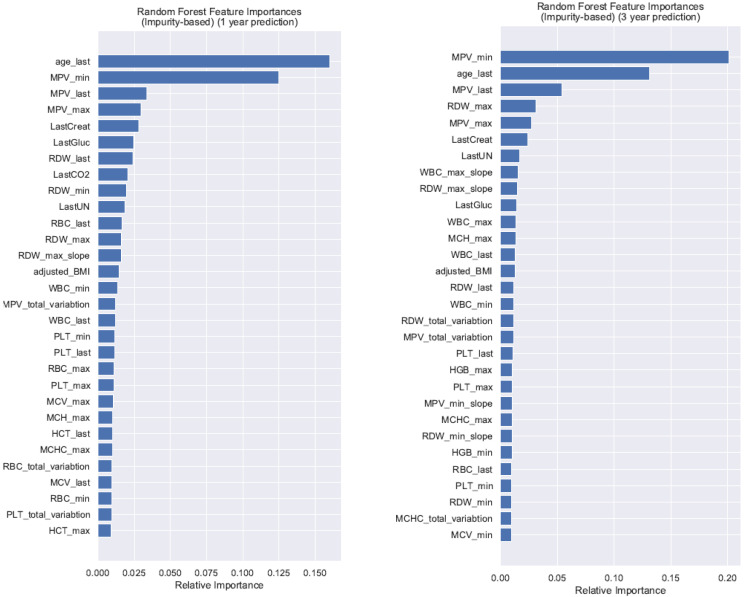
Random Forest model feature importance factors for prediction of GI tract cancers at 1 year and 3 years. Abbreviations: BMI = Body Mass Index; Creat = Creatinine; Gluc = Glucose; HCT = Hematocrit; HGB = Hemoglobin; MCH = Mean Corpuscular Hemoglobin; MCHC = Mean Corpuscular Hemoglobin Concentration; MCV = Mean Corpuscular Volume; MPV = Mean Platelet Volume; PLT = Platelet; RBC = Red Blood Cell; RDW = Red Cell Distribution Width; UN = Blood Urea Nitrogen; WBC = White Blood Count.

**Table 1 cancers-15-01399-t001:** Baseline Population Demographics for the entire cohort.

Total Cohort	*n* = 148,158	
**Sex**		
Male	56,163	37.9%
Female	91,995	62.1%
**Age**		
Mean	49.4 ± 17.3	
Median (IQR)	50 (35–62)	
Range	18–104	
**Race**	Frequency	(%)
Caucasian	120,385	81.3
African American	15,510	10.5
Asian	6795	4.6
Native American	463	0.3
Native Hawaiian/Pacific Islander	76	0.1
Other	3070	2.1
Unknown	1859	1.3

**Table 2 cancers-15-01399-t002:** Results of prediction models. (AUROC = Area Under the Receiver Operating Curve, PPV = Positive Predictive Value, NPV = Negative Predictive Value).

Model Tested	Time	AUROC (95% CI)	Optimal Cutoff	PPV	NPV	Sensitivity	Specificity	F-Score	Brier Score
Logistic reg. (Single Timepoint)	6 month	0.697 (0.679, 0.715)	0.009	0.014	0.996	0.603	0.690	0.027	0.007
	1 year	0.693 (0.675, 0.710)	0.494	0.014	0.996	0.682	0.611	0.027	0.224
	3 years	0.683 (0.665, 0.701)	0.501	0.011	0.996	0.652	0.635	0.022	0.222
	5 years	0.703 (0.686, 0.720)	0.491	0.012	0.996	0.620	0.664	0.024	0.213
Logistic reg. (Longitudinal)	6 month	0.711 (0.691, 0.731)	0.008	0.014	0.996	0.665	0.634	0.027	0.008
	1 year	0.705 (0.689, 0.722)	0.007	0.014	0.997	0.737	0.600	0.027	0.008
	3 years	0.735 (0.713, 0.757)	0.472	0.014	0.997	0.733	0.653	0.027	0.205
	5 years	0.672 (0.653, 0.691)	0.447	0.010	0.997	0.694	0.581	0.020	0.208
Random Forest (Longitudinal)	6 month	0.713 (0.689, 0.737)	0.315	0.015	0.996	0.629	0.671	0.029	0.092
	1 year	0.722 (0.705, 0.739)	0.381	0.015	0.996	0.677	0.660	0.029	0.134
	3 years	0.750 (0.729, 0.771)	0.368	0.015	0.997	0.689	0.695	0.029	0.116
	5 years	0.660 (0.637, 0.682)	0.323	0.011	0.996	0.561	0.697	0.022	0.097

**Table 3 cancers-15-01399-t003:** Performance of models in the test datasets.

Model Tested	Time	Test Set Size	Prediction Success	Prediction Failure	True Positive	False Positive	True Negative	False Negative
Logistic reg. (Single Timepoint)	6 month	21,798	15,018	6780	94	6718	14,924	62
	1 year	27,357	16,735	10,622	152	10,551	16,583	71
	3 years	22,065	14,018	8047	92	7998	13,926	49
	5 years	18,318	12,155	6163	75	6117	12,080	46
Logistic reg. (Longitudinal)	6 month	21,012	13,320	7692	111	7636	13,209	56
	1 year	25,536	15,342	10,194	146	10,142	15,196	52
	3 years	20,187	13,197	6990	99	6954	13,098	36
	5 years	16,100	9368	6732	68	6702	9300	30
Random Forest (Longitudinal)	6 month	21,012	14,101	6911	105	6849	13,996	62
	1 year	25,536	16,859	8677	134	8613	16,725	64
	3 years	20,187	14,025	6162	93	6120	13,932	42
	5 years	16,100	11,215	4885	55	4842	11,160	43

## Data Availability

The datasets generated and/or analyzed during the current study are not publicly available due to potentially identifiable patient information. A limited aggregate dataset is available from the corresponding author on reasonable request.

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
