# Peer review of "Prediction of Gastrointestinal Tract Cancers Using Longitudinal Electronic Health Record Data"

_cancers, 2023, doi:10.3390/cancers15051399_

Round 1
Reviewer 1 Report
The manuscript described a study using longitudinal EHR data (clinical parameters that we use routinely in clinical practice) to predict gastrointestinal cancers. Of interest, different computational models including a machine learning method were compared. The article used complicated computational models and statistical methods which are well beyond my knowledge. Therefore, I am not qualified to judge the methods. I recommend including a statistician to examine the methods.
I am only qualified to judge the originality and significance of the article. The idea is not entirely novel. There have been similar publications on gastric cancer and CRC. The novelty of the article lies on the comparison between different computational models and demonstration of weights of various clinical parameters. Although these models cannot replace existing CRC screening programs, it has the potential to complement the current screening programs with further adjustment and validation. It adds to the literatures of similar studies. I recommend publishing the paper based on the above reasoning (assuming the methods passed the examination by a statistician).
Author Response
Thank you. Our study team includes a statistician, Ji Zhu, PhD, Professor of Statistics at the University of Michigan. We have addressed the other reviewer’s questions re: methods for inclusion criteria and model selection criteria.
Reviewer 2 Report
This nice and well-presented paper describes an machine-learning based approach to predict incidence of GI-tract cancers through subtle changes in complete blood count. Reported AUC up to 0.750 suggests that these results have some true impact.
There are few issues to be fixed prior publishing the paper.
Major problem with the setting is that the authors use a population referred to a large hospital as training cohort. Since the aim is to predict occurrence of cancer, to population should be as cancer-free as possible at baseline. From exclusion criteria in section #2 I cannot find this. Authors should exclude all patients with history of cancer based on ICD-codes and the Michigan cancer registry. ALso, if possible, authors should exclude patients that are referred to the hospital due to suspicion of GI cancer, such as those diagnosed with IDA and referred to colonoscopy.
Overall, the paper should be shortened in each section.
I also noticed that abstract and simple summary, do not necessarily explain what is actually found in the study. THe rationale of the study is the transient bleeding caused by occult cancer, explained in the summary but NOT in the abstract.
How does the model incorporate the interval between the two CBC measurement? I.e. authors required up to a 2-year interval. Short-term changes (2-month?) in the CBC could be more precise concerning occurrence of cancer?
Minor remarks:
Abstract:
“when treatments options are more limited. “ The statement is actually controversial, since there are actually more palliative treatment modalities available than curative. Recommend rephrasing.
Enrollment was 2004-2013, and follow up until 2018, please clarify in the abstract
Do not use abbreviations in the abstract
INTRODUCTION. Which screening methods are used in Michigan?
RESULTS: Tables 1 and 2 repeat what is stated in the text, suggest removal
FIG2 most interesting, abbreviations are missing from the figure legend. For example, MPV appears to be most influential in addition to age, referring to mean platelet volume, explainened only in 3.4.
CONCLUSIONS: References to COVID, even though true, seem to be out of scope of the current study
Round 2
Reviewer 2 Report
The authors have sufficiently responded to the previous criticism. Still, my concern is, that the major issue that I raised earlier "Major problem with the setting is that the authors use a population referred to a large hospital as training cohort." remains unclear in the text. Many readers outside the US do not necessarily understand how the referral system in Michigan workds. The authors have nicely and sufficiently explained this in their reply, but the actual changes in the text are minor. Suggest adding a couple of sentences similiar to those in the responses to the discussion section.
Another point, table 1 is fine, but I would still suggest table 2 as text and/or supplementary.
Author Response
The authors have sufficiently responded to the previous criticism. Still, my concern is, that the major issue that I raised earlier "Major problem with the setting is that the authors use a population referred to a large hospital as training cohort." remains unclear in the text. Many readers outside the US do not necessarily understand how the referral system in Michigan works. The authors have nicely and sufficiently explained this in their reply, but the actual changes in the text are minor. Suggest adding a couple of sentences similar to those in the responses to the discussion section.
- Thank you for this comment. We have added the requested clarifications to the discussion section, which help further clarify the population:
“The eligible population included all patients receiving care at Michigan Medicine, which includes patients seen by Michigan Medicine primary care providers and specialists. We further narrowed our cohort to individuals with longitudinal follow-up within the Michigan Medicine health system by requiring at least 2 CBCs over two years. To maximize our eligible cohort, we did not exclude individuals based on the type of provider(s) seen at Michigan Medicine or other exclusions.”
Another point, table 1 is fine, but I would still suggest table 2 as text and/or supplementary.
- Thank you for the suggestion. We have removed Table 2, and summarized the remaining data in a sentence in 3.1 results:
“Within the cohort, we identified 1,025 GI tract cancers during the study period: the majority were CRCs (53.5%, n = 548/1025), followed by gastric cancer (16.6%, n = 170/1025), esophageal cancers (12.5%, n = 128/1025), anal cancers (8.6%, n = 88/1025), small bowel cancers (7.7%, n = 79/1025), and not otherwise specified within the GI tract (1.2%, n = 12/1025).”
- Subsequent Table numbering has been adjusted accordingly in the manuscript.
